# Can a Win–Win Situation of Economy and Environment Be Achieved in Cities by the Government's Environmental Regulations?

**Xinfei Li, Baodong Cheng \*, Qiling Hong** [1] **and Chang Xu**

School of Economics and Management, Beijing Forestry University, Beijing 100083, China; lxfbjfu@163.com (X.L.); m18750051658@163.com (Q.H.); xuchang@bjfu.edu.cn (C.X.)

\* Correspondence: baodongcheng@163.com

**Abstract:** Based on the panel data of 216 prefecture-level cities in China from 2003 to 2016, this study selected five emission-reduction indicators (industrial $SO_2$ removal rate, soot removal rate, comprehensive utilization rate of industrial solid waste, domestic sewage treatment rate, and harmless treatment of domestic waste rate) to quantify the intensity of urban environmental regulations. Based on the intensity of environmental regulations, the authors further studied the impact of environmental regulations on economic quality (green total factor productivity) and environmental quality (PM2.5). The test results showed that the impact of environmental regulation on PM2.5 is a U-type change that first declines and then rises, while the impact of the implementation of environmental regulation on green total factor productivity is an inverted U-shaped change, which first increases and then decreases. On the one hand, appropriate environmental regulations are conducive to improving environmental quality and improving urban green total factor productivity. On the other hand, excessive environmental regulations have not only failed to improve environmental quality, but also have a negative impact on the improvement of economic quality. In addition, there are regional differences in the impact of environmental regulations, so it is necessary to formulate appropriate and local environmental regulatory policies.

**Keywords:** environmental regulation; economic quality; environmental quality; green total factor productivity





## 1. Introduction

The rapid economic development in China has significantly improved people's living standards, yet at the same time it has brought about some negative externalities such as increasing energy consumption and environmental pollution [1,2]. Due to the large differences in resource endowments and economic-development levels between regions in China, the government has implemented fiscal decentralization and environmental decentralization strategies, which make it possible for local governments to reduce environmental regulations in pursuit of economic growth [3]. The result has generated a number of discussions by people from all walks of life wondering whether economic development must be at the cost of environmental resources. Can a path for high-quality economic development be found without pollution or less environmental pollution? Some developed countries would like to ensure that the environmental quality is not affected. On the basis of promoting the economic-development level, one after another, countries have changed the mode of development, gradually eliminated the extensive production mode that pollutes the environment, and tried to use effective environmental regulations for pollution control. China also got started on gaining relevant experience, attaching more attention to the green transformation of economic development, and clearly putting forward interrelated institutional standards for regulating environmental behavior. Governments at all levels

are urged to adopt reasonable environmental policies and improve the ecological protection system to reduce environmental pollution and therefore prevent ecological deterioration.

The most salient manifestation of urban environmental quality is the emission performance of PM2.5 [4,5]. Green total factor productivity (GTFP), as a variable in the traditional measure of economic growth, also has been widely used academically in the field of economic development [6]. Green total productivity refers to the total factor productivity estimated by taking pollutant emissions as unpaid inputs and introducing them into the production function together with capital, labor, and energy inputs [7]. A remarkable shift in China's economy has been emerging, from a stage of rapid growth to a stage of high-quality development. Environmental protection and the promotion of high-quality economic development are two key objectives of local governments, and the effect of environmental regulations is also valued by all parties. Hence, this paper seeks to facilitate efforts to understand the impact of environmental regulations on both goals.

According to existing studies, there are mainly three views on the impact of environmental-regulation intensity on GTFP. The first view is that environmental regulations hinder the improvement of GTFP [8,9]. According to this view, strict environmental regulations can increase the cost of pollution control and inhibit enterprises' R&D (research and development) and innovation activities, thus inhibiting the GTFP [10–14]. For example, Yuan and Zhang [15] measured the GTFP of Chinese manufacturing enterprises and found that pollution control costs caused by environmental regulations had a "crowding-out effect" on R&D investment. Mohamed et al. [16] and Aklin [17] believed that environmental regulations raised the production costs and management costs of enterprises in cities. Excessive environmental regulations made enterprises tend to sacrifice long-term interests or even make wrong decisions in strategic development in order to meet the corresponding environmental standards, which was not conducive to the improvement of GTFP. Through empirical research, Zhang and Wei [18] not only found that investment in environmental regulations and R&D could significantly promote GTFP, but also found that the impact of foreign direct investment and industrial structure change was significantly negative. The second kind of environmental regulations promote the improvement of GTFP [19,20]. This view holds that reasonable environmental regulations can promote technological innovation of enterprises in cities and offset the environmental governance costs of enterprises, and thus improve GTFP [21]. Chen et al. [22] used the directional distance function (DDF) and global Malmquist–Luenberger productivity index to quantify the GTFP, and found that environmental regulations can promote GTFP in China. Compared with technology introduction, independent research and development plays a more significant role in promoting green total factor productivity in low- and medium-pollution industries. In particular, the improvement of environmental-regulation intensity would enable enterprises to increase R&D investment and carry out innovation in the factors of input, energy consumption, energy conservation, and emission reduction to further improve the competitiveness of enterprises and increase their output. This would make up for the decline in corporate profits caused by the increase in environmental governance costs and continuously promote the improvement of green total factor productivity [23–27]. The third view holds that there is a nonlinear relationship between environmental regulations and GTFP [28,29]. Wang et al. [30] found that there was a nonlinear relationship between environmental regulations and GTFP from the empirical result of the panel threshold model. Wang and Shen [9] found that an inverted "U"-shaped relationship was revealed between environmental-regulation intensity and GTFP with three thresholds. The relationship between the two objects was also significantly distinguished among different industries.

In addition, there are also three views on the impact of environmental regulations on environmental quality. First, increasing environmental regulations makes no sense to the improvement of environmental quality, and even negatively affects it. The enhancement of environmental regulations leads to accelerated exploitation and consumption of fossil energy, which in turn leads to the rapid expansion of air-pollution emissions and the aggravation of haze pollution. Environmental regulation has not played its due inhibiting

role, especially at a time when the problem of space spillover caused by haze pollution has not been solved yet. To some extent, it has worsened the environmental pollution situation [31]. Many studies hold the second insight that environmental regulations have a significant positive effect on energy conservation and emission reduction to curb haze pollution. Levinsohn and Petrin [32] used the empirical results with relevant data on the paper industry in the United States interlinked to reveal that high-intensity environmental regulations would reduce the productivity of the paper industry and thus contribute to the prevention and control of haze pollution [33]. Third, the effect of environmental regulations on pollution is uncertain. Environmental regulations would make an indirect difference to smog pollution through their influence on the adjustment of industrial structure, energy use, and urban transit systems, with all these intermediaries further promoting or inhibiting the fog haze pollution in the space [34–36]. Therefore, we could not precisely infer the influence of environmental regulations on GTFP.

There are two defects in the existing research. (1) Different researchers have used different samples in their research. Some researchers have conducted studies at the na-tional level, while others have considered the provincial level, which is an approach that is prone to errors. (2) Few studies have considered whether there are regional differences in the impact of environmental regulations on the economy and environment. Differences in the level of economic development often determine the policy objectives of local governments, industrial structure, and consumer preferences, which will influence the effect of environmental regulations. In order to make up for the deficiencies of existing research, this article has made the following improvements: based on the municipal panel data of 216 cities in China from 2003 to 2016, this paper constructs an econometric model to analyze the impact of different degrees of environmental regulation on economic quality (GTFP) and environmental quality (PM2.5). In addition, this paper analyzes the impact of environmental regulations on economy and environment in different regions, combined with the differences of geographical location and economic development level of coastal and inland cities, and provides policy guidance according to local conditions.

In view of the reality, this paper attempts to answer the following questions: (1) What are the positive and negative impacts of environmental regulations on the local economy and environment? (2) Does the implementation of environmental regulations restrain or promote economic development while promoting the improvement of environmental quality? Is it a win–win situation between the economy and the environment? This is the focus of local governments' environmental regulations and the key focus of this paper. Reasonable answers to the above two questions will not only help to improve the relevant literature research on environmental regulations, but also play a positive role in guiding the Chinese government to make rational use of environmental-regulation policies to promote the high-quality development of local economies and environmental improvement, which also promotes the sustainable development of society to a certain extent, and has important theoretical and practical significance for the economic transformation of Chinese society.

## 2. Theoretical Analysis

On the environmental impact of environmental regulation, there are two main points of view. The first one is that environmental regulations facilitate no positive efforts toward the improvement of environmental quality, but play a negative role, referring to the Green Paradox theory. According to this theory, strengthening the government's environmental regulations may contribute to a rise in the cost of enterprises' environmental settlements. Under the pressure of financial constraint, enterprises are unable to conduct technological innovation for higher output levels. Driven by the attempt at profit maximization, they may increase output, including pollution emissions. Similar conclusions have been reached by numerous foreign scholars. Schou [37] found that environmental-regulation measures were redundant; Lucas [38] took a project introduced by the Mexico City government as the research background, and indicated that there was no sufficient evidence showing that restrictive measures could improve air quality. Oliva [39] took the automobile exhaust

emission of Mexico City as an example, and found that environmental regulations generated a limited effect on reducing automobile exhaust emissions. The second point of view is that a conspicuous positive effect on energy conservation and emission reduction can be tested by environmental regulations through compulsory emission reductions; that is, the appropriate environmental regulations become conducive to urging enterprises to be financially involved in environmental governance and technology innovation, thus reducing the negative impact on the environment. For example, higher environmental standards would eliminate the enterprises with large pollution emissions and without obvious achievements in pollution constraint, and thereby would stimulate the improvement of urban environmental quality [40]. Magat and Viscusi [41] revealed that environmental regulations issued by the U.S. government could cut down the emissions of pulp enterprises by almost 20%. Laplante et al. [42] offered insight that the Canadian government's environmental supervision not only is beneficial to paper companies in reducing air pollutants, but also can force companies to disclose more comprehensive and accurate information related to emissions. An empirical study was conducted by Dasgupta et al. [43] on the data of polluting enterprises in Zhenjiang from 1993 to 1997 that applied the generalized method of moments (GMM), finding that environmental regulations played a notable role in decreasing air pollutants. Dasgupta et al. [44] analyzed the time series relationship between environmental regulations and pollution emission levels, and found that compared with no environmental regulations, pollutant emission levels would be significantly decreased with higher environmental-regulation intensity. Through experiments, Cole et al. [45] concluded that environmental regulations had effective control over the discharge of multiple pollutants, which was conducive to improving environmental quality.

The existing analysis of the impact of environmental regulations on the economy is mainly based on the "Porter hypothesis" and "Constraint hypothesis" without consistency. According to the Porter hypothesis, sound macroscopic environmental regulations can encourage enterprises to innovate technologically, increase output on the basis of reducing emissions, generate the "compensation effect", and eventually achieve a win–win situation between "economic development" and "environmental protection". Enterprises will be required to change the mode of production under moderate environmental regulations, to separate economic growth from environmental pollution, so the win–win situation between the economy and environment can be achieved. Moreover, from the macro perspective, requirements put forward by environmental regulations promote the development of environmental-protection industries and emerging industries with high economic benefits and little environmental impact, and promote the transformation and upgrading of traditional industries, thus accelerating the improvement of economic quality. Existing studies have proposed a similar view that environmental regulations are beneficial to economic development. Taking the paper industry as an example, Boyd and McClelland [46] analyzed the impact of environmental regulatory constraints on productivity, and found that more strict environmental regulatory constraints reduced 2–8% of pollution emissions without lowering productivity, which can be seen as a "win–win situation". Repetto et al. [47] revealed that pollution costs would be reduced by the environmental regulations, and further, the growth of total factor productivity of enterprises can be promoted. By studying the total factor productivity of enterprises in Los Angeles, Berman and Bui [48] found that the total factor productivity of regulated enterprises rose significantly from 1982 to 1992. Using manufacturing data from seven European countries, Franco and Marin [48] analyzed the impact of energy tax on technological innovation and total factor productivity of enterprises, and illustrated that environmental regulations directly promoted the improvement of total factor productivity. In addition to the positive impact of environmental regulations on the boost of industrial total factor productivity in technologically advanced countries, Albrizio et al. [49] also found that environmental regulations brought the temporary increase of total factor productivity to most enterprises with high productivity. Moreover, the "Constraint hypothesis" proposes that environmental regulations will increase enterprise costs, affect the updating of enterprise technologies, and thus affecting the enhancement



of competitiveness [50]. However, Dufour et al. [51] presented that higher environmental-regulation intensity led to a decrease in total factor productivity (TFP) growth, based on the analysis of manufacturing data. Greenstone et al. [52] found that a 2.6% drop in total factor productivity (TFP) in response to strict air-quality environmental regulations and ozone-related environmental regulations also showed an unfavorable impact on total factor productivity (TFP) of enterprises.

The existing insight regarding environmental impact on environmental regulations reflects that external environmental-regulation policies are generally conducive to improving the environment. However, different environmental regulatory policies and regulatory intensities have been adopted by cities of different development levels, and their environmental effects are differentiated. Moreover, on account of disaccord in economic-development levels and policy implementation in different regions, there are also some certain differences in the environmental efficiency of related regulations. Current studies' conclusions on the economic impact of environmental regulations are not exclusive. Both the "Porter hypothesis" and the "Restriction hypothesis" have been supported by relevant studies. Whether environmental regulations can bring about an expected win–win situation for the economy and environment remains to be tested. The authors found that the environmental regulatory variables, pollution indicators, and economic indicators selected by other researchers are varied, and few scholars simultaneously studied the impact of different regulatory methods on economic and environmental effects. Many studies introduced per capita GDP or traditional total factor productivity when estimating economic effects, only considering the input constraints of production factors such as capital and labor. When measuring production efficiency, environment was hardly considered as a constraint, nor were undesired products such as pollution emissions. The economic effect was overestimated.

Based on the above, this article further analyzes the actual economic and environmental effects brought about by environmental regulations, and discusses whether environmental regulations will help China and different regions achieve a win–win situation for the economy and environment. Figure 1 shows the theoretical framework of this paper. Here, we explore mainly from the following aspects, First, we recognize the impact of environmental regulations on the quality of economic growth. A remarkable shift from a high-speed growth stage to a high-quality growth stage of China's economic is ongoing, so the central point of economic growth is high-quality development. The Malmquist–Luenberger (ML) index model is applied to avoid the error of the traditional literature on the measurement of economic quality to the greatest extent, and to quantify the green total factor productivity, including energy consumption and pollution emissions, to represent the quality of economic growth. Second, this paper introduces the quadratic term of environmental regulations to test the nonlinear effects of environmental regulations. Third, due to the uneven economic development of various regions, this study partitions selected prefecture-level cities into coastal and inland regions for later panel data analysis.

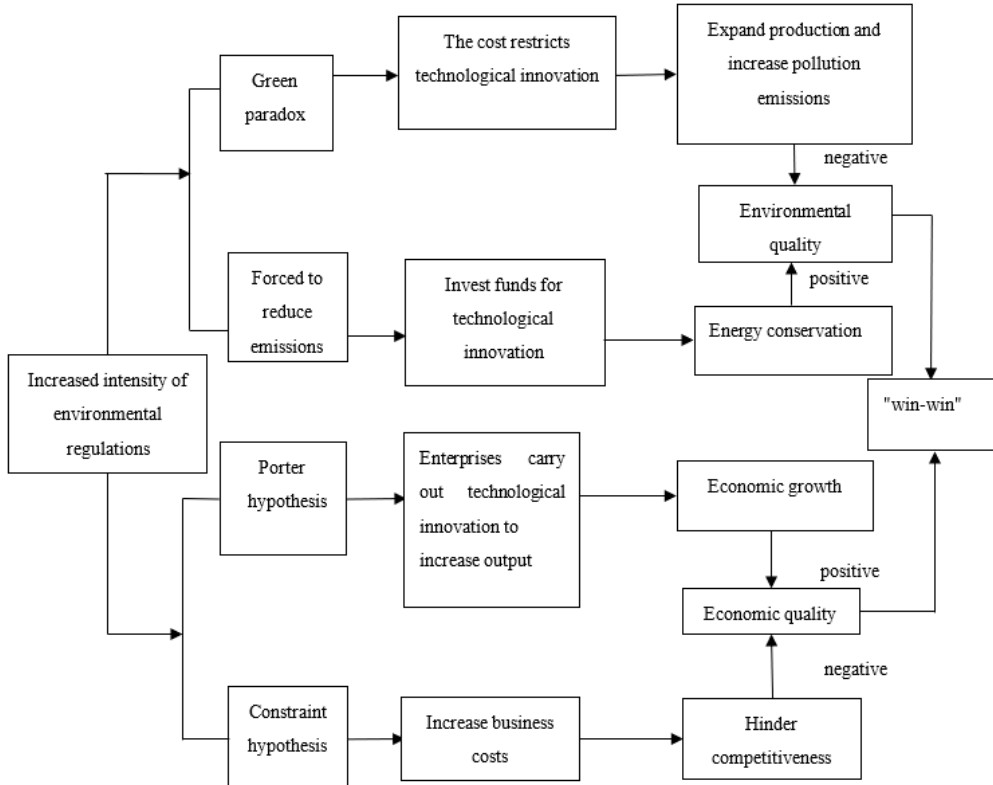

**Figure 1.** The theoretical framework of this study.

## 3. Research Design

### 3.1. Model Specification

With a view toward revealing the accurately quantified impact of environmental regulations on the environment and economy, this paper establishes the following model for analysis. The impacts of different types of environmental regulations on the environment and economy were tested with national, inland, and coastal samples. Models (1) and (3) were to test the linear and nonlinear impact of environmental regulation on environmental quality. $PM25_{i,t}$ represents smog, and was selected as the environmental quality index. Models (2) and (4) were to test the linear and nonlinear impact of environmental regulations on economic quality. $GTFP_{i,t}$ means green total factor productivity, which represents the quality of economic growth. In addition, $CONTROLS_{i,t}$ represents the control variable, and $\varepsilon_{i,t}$ is a random perturbation term.

$$PM25_{i,t} = \alpha_0 + \alpha_1 GER_{i,t} + \alpha_n CONTROLS_{i,t} + \varepsilon_{i,t} \tag{1}$$

$$GTFP_{i,t} = \beta_0 + \beta_1 GER_{i,t} + \beta_n CONTROLS_{i,t} + \varepsilon_{i,t} \tag{2}$$

$$PM25_{i,t} = \alpha_0 + \alpha_1 GER_{i,t} + \alpha_2 GER_{i,t} * GER_{i,t} + \alpha_n CONTROLS_{i,t} + \varepsilon_{i,t} \tag{3}$$

$$GTFP_{i,t} = \beta_0 + \beta_1 GER_{i,t} + \beta_2 GER_{i,t} * GER_{i,t} + \beta_n CONTROLS_{i,t} + \varepsilon_{i,t} \tag{4}$$

### 3.2. Variable Selection

#### 3.2.1. Green Total Factor Productivity (GTFP)

Because the price information for resources and environmental factors is difficult to be acquire, traditional total factor productivity measurements cannot calculate the productivity with resource and environmental constraints. Although price information is not required when calculating the productivity index based on the traditional distance function, it is difficult to calculate the total factor productivity in the presence of "bad"

output (such as $SO_2$ emissions). Chung et al. [53] first proposed the Malmquist–Luenberger (ML) index based on the directional distance function, which can measure the total factor productivity even if there is unfavorable output. The data envelopment analysis (DEA) model for measuring efficiency is classified by the existing research insight as radial and angular, radial and nonangular, nonradial and angular, and nonradial and nonangular. "Radial" means that the input or output should be adjusted by the same proportion when evaluating efficiency, while "angular" means that, when evaluating efficiency, it is necessary to make an input-based (assuming that output remains unchanged) or output-based (assuming that inputs change) DEA selection. Inasmuch as DEA holds the advantage of not requiring assumption of the function form and is able to decompose productivity, many archives basically use radial and oriented DEA to solve the directional distance function. However, radial DEA and angular DEA have explicit shortcomings; that is, when over-input or under-output exists, the radial DEA efficiency measure will overestimate the efficiency of the evaluated object. In addition, the perspective of DEA efficiency measurement ignores a certain aspect of input or output with an inaccurate calculated result of efficiency. In order to combat these two shortcomings, nonradial and nonangular directional distance function-Slack-based measure (SBM) began to be widely used. Based on the existing studies of nonradial and nonangular directional distance function (SBM) and the Malmquist–Luenberger (ML) index, this paper refers to the method applied by Haifeng et al. [54] and adds energy input as a factor using the Malmquist–Luenberger index based on the nonradial SBM directional distance to calculate the dynamic changes of the GTFP of 163 cities in China from 2003 to 2016.

Here, each city is regarded as a decision-making unit (DMU) to construct the production frontier. It is assumed that each DMU uses N kinds of inputs $x = (x_{1,}, x_{N,}) \in R_N^+$, and produces M kinds of expected outputs $y = \left(y_{1,}, y_{M,}\right) \in R_M^+$, accompanied by a Type I unexpected output $b = (b_{1,}, b_{I,}) \in R_I^+$. In each period $t = 1, \cdots, T$, the production possibility set of the $k = 1, \cdots, K$ city is $\left(x^{k,t}, y^{k,t}, b^{k,t}\right)$. Data envelopment analysis (DEA) is applied in the modeling as follows:

$$P^t\left(X^t\right) = \left\{ (y^t, b^t) : \sum_{k=1}^{k} \lambda_k^t y_{km}^t \geq y_{km}^t, \forall m; \sum_{k=1}^{k} \lambda_k^t b_{ki}^t = y_{ki}^t, \forall i; \sum_{k=1}^{k} \lambda_k^t x_{kn}^t \right. \tag{5}$$
$$\left. \leq y_{kn}^t, \forall n; \sum_{k=1}^{k} \lambda_k^t = 1, , y_k^t \geq 0, \forall k; \right\}.$$

This represents the weight of each cross-sectional observation value. $\Sigma_{k=1}^{k} \lambda_k^t = 1$, $y_k^t \geq 0$, $\forall k$ means that the production technology has a variable return to scale.

Each city is considered to be a DMU to construct the productive frontier. The production possibility set of each DMU is presented as $(x, y, b)$. Because the directional distance function not only is in keeping with the relevant properties of the production possibility set, but also reflects the directional properties shown by the output in the production process, the directional distance function is expressed by the following formula:

$$\overrightarrow{D_0}(x, y, b; g) = \sup\{\beta : (y, b) + \beta g \epsilon P(x)\} \tag{6}$$

where $g = (gy, gb)$ is the direction vector used to indicate the direction of output expansion. According to the DEA model, to solve the directional distance function, the following linear programming equation can be set:

$$\overrightarrow{D_0}\left(x^{t,k}, y^{t,k}, b^{t,k}; y^{t,k}, -b^{t,k}\right) = \text{Max}\beta \tag{7}$$

$$\sum_{k=1}^{k} \lambda_k^t y_{km}^t \geq (1+\beta) y_{km}^t, m = 1, 2, \cdots, M$$

$$\text{s.t.} \sum_{k=1}^{k} \lambda_k^t b_{ki}^t = (1-\beta) b_{ki}^t, i = 1, 2, \cdots, I \qquad (8)$$

$$\sum_{k=1}^{k} \lambda_k^t b_{kn}^t \leq x_{kn}^t, n = 1, 2, \cdots, N$$

The ML productivity index is used to represent GTFP based on environmental factors. The ML index from period t to period $t+1$ based on outputs is:

$$ML_t^{t+1} = \left\{ \frac{\left[1 + \overrightarrow{D}_0^t\left(x^t, y^t, b^t; g^{t+1}\right)\right]}{\left[1 + \overrightarrow{D}_0^t\left(x^{t+1}, y^{t+1}, b^{t+1}; g^{t+1}\right)\right]} \times \frac{\left[1 + \overrightarrow{D}_0^{t+1}\left(x^t, y^t, b^t; g^{t+1}\right)\right]}{\left[1 + \overrightarrow{D}_0^{t+1}\left(x^{t+1}, y^{t+1}, b^{t+1}; g^{t+1}\right)\right]} \right\}^{\frac{1}{2}} \qquad (9)$$

The ML index can be further decomposed into the technological progress index (GTC) and the technological efficiency change index (GTEC). The GTC is introduced to measure the shift of the production possibility boundary caused by technological progress, while GTEC is considered to measure the policy and system improvements.

$$ML = GTEC \times GTC \qquad (10)$$

The expression of GTEC is:

$$GTEC_t^{t+1} = \frac{1 + \overrightarrow{D}_0^t\left(x^t, y^t, b^t; g^t\right)}{1 + \overrightarrow{D}_0^t\left(x^{t+1}, y^{t+1}, b^{t+1}; g^{t+1}\right)} \qquad (11)$$

The expression of GTC is:

$$GTC_t^{t+1} = \left\{ \frac{\left[1 + \overrightarrow{D}_0^t\left(x^t, y^t, b^t; g^t\right)\right]}{\left[1 + \overrightarrow{D}_0^t\left(x^t, y^t, b^t; g^t\right)\right]} \times \frac{\left[1 + \overrightarrow{D}_0^{t+1}\left(x^{t+1}, y^{t+1}, b^{t+1}; g^{t+1}\right)\right]}{\left[1 + \overrightarrow{D}_0^t\left(x^{t+1}, y^{t+1}, b^{t+1}; g^{t+1}\right)\right]} \right\}^{\frac{1}{2}} \qquad (12)$$

For the input indicators, relevant variables were selected such as human input, capital input, and energy consumption input [55–57]. For the output indicators, both the maximization of expected outputs (e.g., economic development) and undesired outputs (e.g., environmental pollution) were selected [58,59]. As a constraint on economic development, the economic output was chosen as the expected output index, and the environmental pollution index was set as the undesired output index. Our data comes from the 2003–2016 "City Statistical Yearbook".

ML > 1 indicates the growth of GTFP from period t to t+1; ML < 1 indicates the decline of GTFP from period t to t+1; and ML = 1 indicates that GTFP is in a stable state. GTFP is obtained through the ML index. Specifically, the ML index represents the growth rate of GTFP, which is a dynamic indicator. For example, Beijing's ML index in 2006 was 1.0791, which means that Beijing's GTFP in 2006 was 1.0791 times that of 2005; that is, an increase of 7.91% over 2005, and the same is applicable for the decomposition factors. Since the GTFP growth rate (ML index) and the decomposition items calculated by the ML index model are both dynamic chain growth indicators, they reflect the improvement of the chain. In order to reasonably reflect the quality of economic growth in the current year, we selected 2003 as the base period to convert the chain growth rate index of GTFP into a fixed rate improvement one; that is, we assumed that the environmental total factor productivity GTFP in 2003 was 1 and the GTFP growth rate was multiplied by the index, meaning that the GTFP in 2004 was the GTFP in 2003 multiplied by the ML index in 2004, the GTFP in 2005 was the GTFP in 2004 multiplied by the ML index in 2005, and so on. The GTFP of the corresponding year was obtained for empirical analysis.

### 3.2.2. Other Variables

PM2.5 was selected to quantify the environmental quality. The lower the levels of PM2.5, the more preferable the environmental quality. China's Department of Ecology and Environment only recently began to monitor and disclose PM2.5 concentration data for all cities. Raster data of global PM2.5 concentration from 2003 to 2016 provided by the Center for Social and Economic Data and Application of Columbia University was applied as PM2.5 concentration data in this paper [60]. Compared with the ground field test data, this kind of data comes from satellite monitoring and is identified as nonpoint source data, which can reveal the change in PM2.5 concentration in a region more comprehensively and accurately.

The explanatory variable is environmental regulatory intensity (GER). At present, the environmental regulation intensity is calculated mainly from the perspectives of environmental capital investment, industrial wastewater discharged up to the standard rate, and the $SO_2$ removal rate. Based on the five indexes of industrial $SO_2$ removal rate, industrial COD removal rate, industrial solid-waste comprehensive utilization rate, domestic sewage treatment rate, and domestic garbage harmless treatment rate, provided by the China Urban Statistics Yearbook over the years, this study obtained the municipal unit environmental regulation intensity index by standardization to gain its entropy value.

To make the econometric model more robust and reduce the estimation error caused by missing variables, control variables and their measurement methods were selected; these are shown in Table 1.

**Table 1.** Variable interpretation.

| Classification | Name | Interpretation | Symbol | References |
|---|---|---|---|---|
| Explained variable | Economic quality | Green total factor productivity, Malmquist–Luenberger exponent calculation based on nonradial SBM directional distance | GTFP | [61,62] |
| | Environmental quality | PM2.5 concentration data is based on the grid data of global PM2.5 concentration from 2003 to 2016 provided by the Center for Social and Economic Data and Application of Columbia University | PM2.5 | [60] |
| Explanatory variable | Environmental-regulation intensity | The intensity of environmental regulations is calculated by entropy weight method through the five single indexes of industrial $SO_2$ removal rate, smoke and dust removal rate, comprehensive utilization rate of industrial solid waste, domestic sewage treatment rate, and harmless treatment rate of domestic garbage. | GEV | [45,63] |
| Control variable | Level of urban development | GDP growth rate = (GDP of the previous year–GDP of the current year)/GDP of the previous year | GDP | [5] |
| | Industrial structure | Added value of tertiary industry/added value of secondary industry | IS | [5,64] |
| | Opening up | Total industrial output value of foreign-invested enterprises(CNY 10,000)/Gross regional Product (CNY 10,000) | FDI | [65,66] |
| | Informatization | Annual electricity consumption/Year-end total population (10,000 KW/person) | TEL | [67] |
| | Infrastructure | Urban road area per capita | ROD | [68] |
| | Educational level | The natural logarithm of education expenditure ( CNY 10,000) | ED | [69] |
| | Research and development | The natural logarithm of research and development expenditure ( CNY 10,000) | RD | [70,71] |

### 3.3. Data Description

The main research objects of this paper were the cities at prefecture level and above in China. Due to the continuity of data and the availability of variables, the final sample contained 216 cities at the prefecture level and above. The time span was from 2003 to 2016, with a total of 3024 sample values, amounting to 14 years. Among these cities, there were 126 inland cities and 90 coastal cities. Not only important urban agglomerations were included, but also cities in underdeveloped areas. The selected variables were all representative. The data in this paper were collected mainly from the "China Urban Statistical Yearbook" (2004–2017) and the "China Regional Economic Statistical Yearbook" (2004–2017). Foreign direct investment involves exchange rate conversion, which needs converting by the annual average exchange rate of RMB in the China Statistical Yearbook. The estimated sample of this paper was the panel data of 216 cities in China from 2003 to 2016. As the panel regression method was divided into fixed effect and random effect, the Hausmann test first was applied to test the model. All the models in this paper rejected the original hypothesis that "random effect is more effective than fixed effect", so the fixed-effect model was used for all the estimates in this paper.

### 3.4. Correlation Analysis

Table 2 shows the correlation coefficient matrix of variables in the model, and indicates that there was no significant autocorrelation between independent variables. As can be seen from the table, the correlation coefficient between environmental regulation intensity (GEV) and environmental quality (PM2.5) was 0.26, which was significant at 1%, indicating that environmental regulations were not conducive to the mitigation of the current situation of urban haze. It conforms to the hypothesis of the Green Paradox. The correlation coefficient between environmental regulation intensity (GEV) and green total factor productivity (GTFP) was 0.16, also significant at 1%, showing that environmental regulations had a negative impact on green total factor productivity (GTFP). It conformed to the Constraint hypothesis in that environmental regulations will hinder technological innovation and economic growth. Furthermore, the correlation coefficient between each control variable was small, and the variance inflation factor (VIF) was far less than the critical value of 10, so there was no serious multicollinearity problem between variables.

**Table 2.** Correlation analysis.

| | GTFP | PM2.5 | GEV | GDP | IS | FDI | TEL | ROD | ED | RD |
|---|---|---|---|---|---|---|---|---|---|---|
| Mean value | 1.01 | 38.27 | 0.67 | 0.12 | 0.91 | 0.18 | 0.57 | 11.27 | 11.33 | 8.57 |
| GTFP | 1 | | | | | | | | | |
| PM2.5 | 0.03 * | 1 | | | | | | | | |
| GEV | 0.16 *** | 0.26 *** | 1 | | | | | | | |
| GDP | −0.12 *** | −0.01 | −0.11 *** | 1 | | | | | | |
| IS | 0.10 *** | −0.09 *** | 0.13 *** | −0.09 *** | 1 | | | | | |
| FDI | 0.04 ** | 0.23 *** | 0.20 *** | 0.03 * | −0.03 | 1 | | | | |
| TEL | 0.07 *** | −0.08 *** | 0.19 *** | −0.08 *** | −0.11 *** | 0.16 *** | 1 | | | |
| ROD | 0.11 *** | 0.15 *** | 0.36 *** | −0.10 *** | −0.02* | 0.33 *** | 0.46 *** | 1 | | |
| ED | 0.19 *** | 0.20 *** | 0.55 *** | −0.11 *** | 0.22 *** | 0.37 *** | 0.25 *** | 0.33 *** | 1 | |
| RD | 0.18 *** | 0.22 *** | 0.56 *** | −0.11 *** | 0.18 *** | 0.38 *** | 0.29 *** | 0.39 *** | 0.91 *** | 1 |

Note: The values in brackets are t-statistic values. ***, ** and * mean that the values were shown to be significant at the levels of 1%, 5%, and 10%, respectively.

## 4. Empirical Analysis

### 4.1. Impact of Environmental Regulation on Environmental Quality and Economic Quality

#### 4.1.1. Environmental Quality (PM2.5)

The average PM2.5 index of 216 cities in China from 2003 to 2016 is shown in Figure 2. It can be seen from Figure 2 that fluctuations were present in the downward trend of the overall PM2.5 index during 2003–2016. However, the PM2.5 index showed an upward trend from 2003 to 2007 before declining in 2008. A short-term recovery in 2013 could not retard the downward trend as a whole, which was closely associated with the Chinese government's emphasis on energy conservation and emissions reduction. It is noteworthy that the PM2.5 index in coastal areas appeared higher than the national average, while the haze concentration in inland areas was lower than the national average value. This can be explained by the relatively developed economy in coastal areas, especially the high concentration of gases emitted by manufacturing factories and urban automobile exhaust. Therefore, the pollution was more serious than that in other regions.

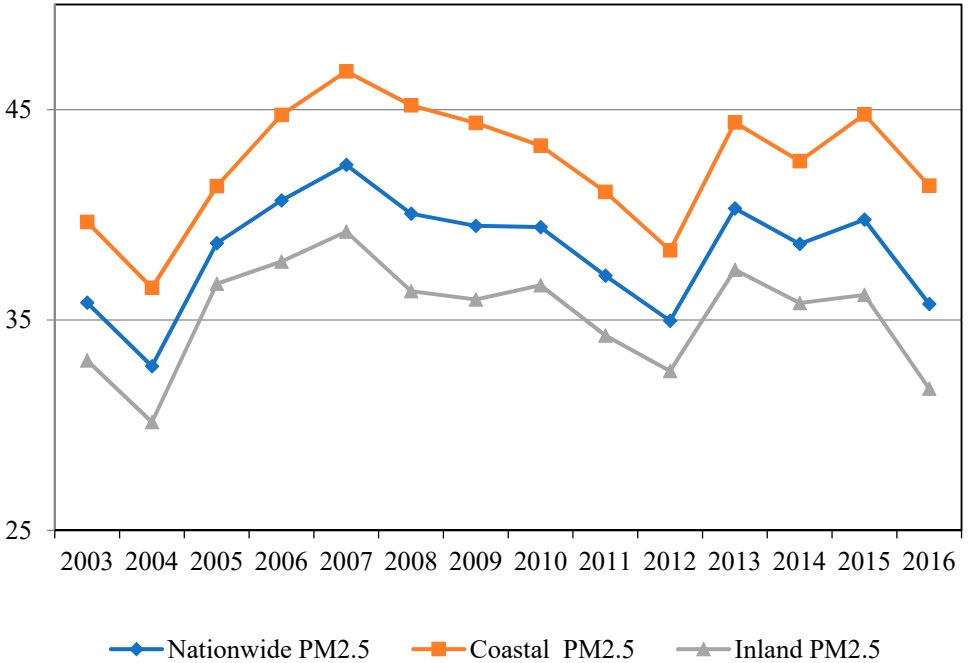

**Figure 2.** PM2.5 index in the national, coastal, and inland areas.

The regression results of environmental regulations on PM2.5 are shown in Table 3. The empirical results of the immediate impact of environmental regulations on environmental quality (Model 1) revealed that, nationally speaking, environmental regulations can significantly expedite the reduction of PM2.5 and alleviate the serious haze situation in the country. Specifically, the impact coefficient of environmental regulations (GEV) on PM2.5 was 1.809, which was significant at a 10% level. As for the control variables, industrial structure (IS) had a significantly positive impact on PM2.5, which means that the greater the ratio of the added value of the tertiary industry to the added value of the secondary industry, the higher the PM2.5 index value will be. Education input (ED) and research and development (RD) had a remarkably negative impact on PM2.5. Conclusions can be made that education and research and development help to alleviate environmental quality and reduce urban haze to a certain extent. The quadratic term of environmental regulations (GEV2) was introduced in Model 2 to test the nonlinear influence of environmental regulations on environmental quality, and the results for Model 2 illustrated that the impact of environmental regulations on environmental quality had an obvious U-shaped feature. The critical point of the second-order type was up to $10.10/(2 \times 6.783) = 0.745$; that is, the quadratic environmental regulations had a certain positive impact on PM2.5, and

were significant at the level of 5%. This further elucidates that excessive environmental regulations are not conducive to improving environmental quality, but will increase the PM2.5 index value, resulting in a higher probability of occurrence of urban haze.

**Table 3.** The regression results for environmental regulations on economic quality and environmental quality.

| Variable | PM2.5 | | GFTP | |
|---|---|---|---|---|
| | **Model 1** | **Model 2** | **Model 3** | **Model 4** |
| GEV | −10.10 ** | −1.809 * | −0.117 *** | 0.391 *** |
| | (4.221) | (0.935) | (0.0336) | (0.151) |
| GEV2 | 6.783 ** | | | −0.416 *** |
| | (3.367) | | | (0.121) |
| GDP | −1.125 | −1.092 | −0.0888 *** | −0.0868 *** |
| | (0.766) | (0.766) | (0.0275) | (0.0274) |
| IS | 0.604* | 0.625 ** | 0.0298 *** | 0.0311 *** |
| | (0.315) | (0.315) | (0.011) | (0.011) |
| FDI | 0.450 | 0.399 | 0.0478 | 0.0447 |
| | (0.859) | (0.859) | (0.0308) | (0.0308) |
| TEL | 0.2080 | 0.220 | −0.0069 | −0.006 |
| | (0.2960) | (0.296) | (0.0106) | (0.0106) |
| ROD | −0.00912 | −0.0110 | 0.0001 | −0.0002 |
| | (0.0193) | (0.0193) | (0.0007) | (0.0007) |
| ED | −0.903 *** | −0.916 *** | −0.005 | −0.006 |
| | (0.293) | (0.293) | (0.01) | (0.0105) |
| RD | −0.636 *** | −0.593 *** | 0.001 | 0.004 |
| | (0.293) | (0.138) | (0.005) | (0.005) |
| Year | Control | Control | Control | Control |
| Reign | Control | Control | Control | Control |
| Constant | 52.8 *** | 50.27 *** | 1.060 *** | 0.905 *** |
| | (3.843) | (3.633) | (0.13) | (0.138) |
| Observations | 3024 | 3024 | 3024 | 3024 |
| R2 | 0.315 | 0.314 | 0.140 | 0.143 |

Note: the values in brackets are t-statistic values. ***, **, and * mean that the values were shown to be significant at the levels of 1%, 5%, and 10%, respectively.

### 4.1.2. Economic Quality (GTFP)

The Malmquist–Luenberger (ML) values of 216 municipal units from 2003 to 2016 were averaged in this study to obtain the average value of ML nationwide. The Malmquist–Luenberger (ML) index is the annual growth rate of green total factor productivity (GTFP). When the ML index is greater than 1 (red dotted line), a greater GTFP value of the year is obtained compared to the previous year and vice versa; the value declines compared with the previous year, which also directly affects the trend of GTFP. As can be seen in Figure 3, the ML value of 216 municipal units in China presented an rising tendency. It revealed that in recent years, though the country and government departments have considered resource consumption and pollution emissions constraint, more attention to China's economic quality is still needed, and there was no situation in which economic growth was at the expense of resources and environment. In terms of different regions, the ML index value for coastal areas was higher than that of inland areas. The main promising reason that may be posited is that the coastal areas realized better economic development, had the economic capacity and full motivation to facilitate technological innovation promoting industrial transformation and earlier upgrading, and thereby led the development of national technological progress. At the same time, the insufficient marketization level, low management level, and unfavorable resource allocation efficiency of the inland areas obviously restricted the improvement of technical efficiency.

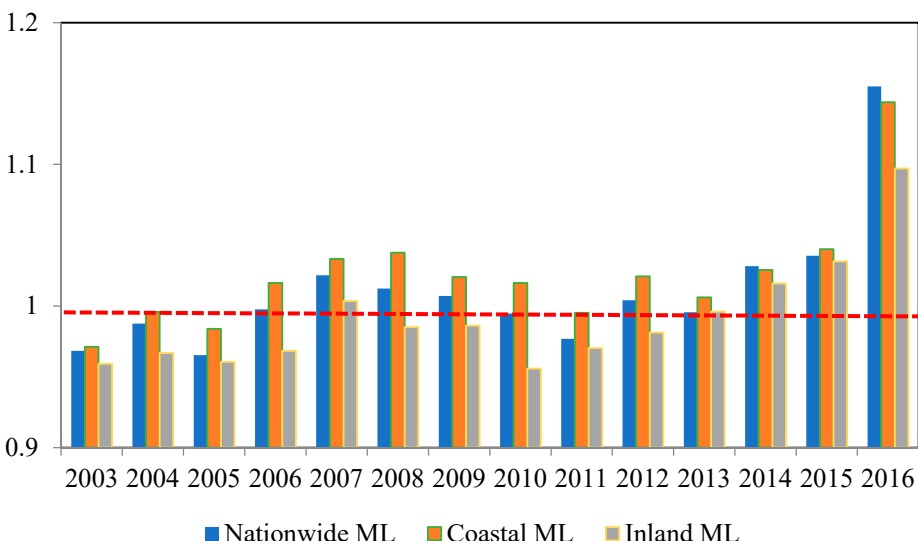

**Figure 3.** Malmquist–Luenberger (ML) index for the national, coastal, and inland areas.

Based on the above, we further explored the impact of environmental regulations on green total factor productivity through empirical analysis (as shown in Table 3). The regression results of Model 3 indicated that environmental regulations (GEV) brought about a significant negative impact on green total factor productivity (GTFP) at the level of 1%, with an impact coefficient of −0.117; that is, environmental regulations (GEV) will obstruct the increase of urban green total factors in productivity. Model 4 introduced the quadratic of environmental regulations on the basis of Model 3, further assessing the nonlinear effect of environmental regulations on green total factor productivity. Model 4 underpinned that from the national perspective, the impact of environmental regulations on green total factor productivity appeared to have obvious nonlinear features. In particular, the quadratic power of environmental regulations showed a negative impact on green total factor productivity, significant at a level of 1%. A positive effect on the green total factor productivity was assessed at a significance level of 1%, referring to the inverted U-shaped characteristic. As it can be concluded that a certain degree of environmental regulation had a positive effect on green total factor productivity, when environmental regulation was greater than the critical point, after $0.391/(2 \times 0.416) = 0.47$, environmental regulation had major negative effects on green total factor productivity. The results emphasized that excessive environmental regulations facilitate no efforts toward promoting green factor productivity, and tend to make it worse. Among the control variables, the regression results of Model 3 and Model 4 were basically consistent with each other. The growth rate of gross regional product also was reported to have a negative impact on green total factor productivity (GTFP), surprisingly revealing that excessive GDP growth may not necessarily promote the increase of green total factor productivity (GTFP). However, the industrial structure (IS) has salient upsides to the green total factor productivity (GTFP), which means the larger the tertiary industry's output value is, the higher the economic quality will be.

*4.2. Research by Region*

Based on insights from this paper, 216 cities were classified by region, namely coastal regions and inland regions, to critically discuss whether there were differences driven by regions regarding effects of environmental regulations on the development of urban environmental quality and economic quality. Therefore, we defined "coastal city" as "coastal city level or above", which was also in line with the definition of "coastal city" in the China Marine Statistical Yearbook. According to this publication, there are 126 inland cities and 90 coastal cities.

The regional regression results for environmental regulations on quality at both the economic and environmental scales are shown in Table 4, which outlines that the PM2.5

index value could be significantly reduced by environmental regulations in coastal areas, and the constraint effectively curbed urban haze, with an impact coefficient up to −2.56. Though a negative impact on PM2.5 appeared again in environmental regulations implemented in inland areas, it was inadequately significant, and the coefficient was smaller compared with that of coastal areas. Among selected control variables, there was a significantly positive correlation between foreign direct investment (FDI) and PM2.5 in coastal cities, while information technology (TEL), education investment (ED), and science and technology investment (RD) showed notably negative correlations with PM2.5. Referring to inland cities, a significantly negative impact on PM2.5 was revealed regarding education investment (ED) and science and technology (RD) input of the control variables. However, in terms of the impact coefficient, education input (ED) and science and technology input (RD) had a greater impact on coastal areas.

**Table 4.** The influence of environmental regulations on environmental quality (PM2.5) and economic quality (GTFP) under regional conditions.

| Variable | PM2.5 | | | | GTFP | | | |
|---|---|---|---|---|---|---|---|---|
| | Coastal | | Inland | | Coastal | | Inland | |
| | Model 5 | Model 6 | Model 7 | Model 8 | Model 9 | Model 10 | Model 11 | Model 12 |
| GEV2 | | 0.0194 | | 7.457 | | −0.294 ** | | −0.388 ** |
| | | (5.164) | | (4.691) | | (0.131) | | (0.193) |
| GEV | −2.560 * | −2.585 | −1.502 | −10.41 * | −0.126 *** | 0.251 * | −0.136 *** | 0.327 * |
| | (1.470) | (6.794) | (1.223) | (5.732) | (0.0373) | (0.172) | (0.0504) | (0.236) |
| GDP | −0.590 | −0.590 | −3.490 | −3.556 | −0.0875 *** | −0.0862 *** | −0.105 | −0.102 |
| | (0.763) | (0.764) | (2.573) | (2.572) | (0.0193) | (0.0193) | (0.106) | (0.106) |
| IS | 0.618 | 0.618 | 0.663 | 0.688 * | 0.0468 *** | 0.0487 *** | 0.0271 | 0.0258 |
| | (0.518) | (0.519) | (0.414) | (0.414) | (0.0131) | (0.0131) | (0.0171) | (0.0171) |
| FDI | 2.424 ** | 2.424 ** | −2.399 | −2.450 | 0.0433 * | 0.0391 | 0.0128 | 0.0154 |
| | (1.028) | (1.031) | (1.502) | (1.502) | (0.0260) | (0.0261) | (0.0619) | (0.0619) |
| TEL | −2.166 *** | −2.166 *** | 0.427 | 0.418 | −0.0266 | −0.0262 | −0.00175 | −0.0013 |
| | (0.823) | (0.823) | (0.327) | (0.327) | (0.0209) | (0.0208) | (0.0135) | (0.0135) |
| ROD | −0.0057 | −0.0057 | 0.0234 | 0.0237 | 0.00004 | −0.0001 | −0.001 | −0.001 |
| | (0.0237) | (0.0238) | (0.0314) | (0.0314) | (0.0006) | (0.001) | (0.0013) | (0.0013) |
| ED | −0.826 * | −0.826 * | −0.793 ** | −0.797 ** | 0.00249 | 0.002 | −0.0105 | −0.0103 |
| | (0.498) | (0.499) | (0.370) | (0.370) | (0.0126) | (0.0126) | (0.0153) | (0.0152) |
| RD | −0.788 *** | −0.788 *** | −0.506 *** | −0.552 *** | 0.00198 | 0.004 | −0.0003 | 0.002 |
| | (0.220) | (0.223) | (0.179) | (0.182) | (0.00557) | (0.006) | (0.007) | (0.007) |
| Year | Control | Control | Control | Control | Control | Control | Year | Control |
| Region | Control | Control | Control | Control | Control | Control | Region | Control |
| Constant | 55.29 *** | 55.30 *** | 44.10 *** | 46.88 *** | 0.980 *** | 0.856 *** | 1.128 *** | 0.983 *** |
| | (4.827) | (5.301) | (3.520) | (3.929) | (0.122) | (0.134) | (0.145) | (0.162) |
| Observations | 1260 | 1260 | 1764 | 1764 | 1260 | 1260 | 1764 | 1764 |
| R2 | 0.399 | 0.399 | 0.294 | 0.295 | 0.190 | 0.193 | 0.144 | 0.146 |

Note: The values in brackets are t-statistic values. ***, **, and * mean the values were shown to be significant at the levels of 1%, 5%, and 10% respectively.

Considering green total factor productivity (GTFP), environmental regulations had a saliently negative impact in both coastal and inland areas, and both were significant at the 1% level. Specifically, the impact coefficient on inland areas was larger (−0.136), while that on coastal areas was smaller, up to −0.126. Regulations' impact on coastal areas was more significant for control variables. The GDP growth rate generated a remarkably negative impact on green total factor productivity (GTFP). Industrial structure (IS) and foreign investment (FDI) brought significant upsides to green total factor productivity (GTFP). It can be articulated that increasing the development of tertiary industry and improving the level of foreign-invested enterprises in coastal areas were conducive to hoisting green total factor productivity (GTFP).

The relationship between environmental regulation intensity and environmental quality (PM2.5) was U-shaped, and was significantly inverted when describing the correlation

between environmental regulation intensity and green total factor productivity (GTFP) according to the nationwide results. For sake of testing whether these two nonlinearities were driven by regional differences, this paper introduced the quadratic term of environmental regulation (GEV2) from the regional level to assess the nonlinear influence of environmental regulation on PM2.5 and green total factor productivity. Empirical results showed that there was no distinguished U-shaped relationship between environmental regulation and PM2.5 in coastal or inland areas. However, there was an inverted U-shaped relationship between environmental regulation and green total factor productivity (the critical point in coastal areas was marked as $0.251/(2 \times 0.294) = 0.427$, and the critical point in inland areas was $0.327/(2 \times 0.388) = 0.421$). The critical point in inland areas was reached earlier, and it confirmed that excessive environmental regulations had an inhibitory effect on the development of GTFP in both coastal and inland areas, with a higher impact coefficient for inland areas.

### 4.3. Robustness Tests

Drawing on the work of Tao et al. [72], environmental regulation efficiency (ERE) was selected by us to replace the environmental regulation variables in order to further test the robustness. The regression results were basically consistent with the above, which could prove the robustness of the results (as shown in Table 5).

**Table 5.** Robustness test of environmental regulations.

| Variable | PM2.5 | | GTFP | |
|---|---|---|---|---|
| | **Model 13** | **Model 14** | **Model 15** | **Model 16** |
| ERE | −3.296 *** | −6.669 *** | −0.088 *** | 0.204 *** |
| | (0.308) | (0.639) | (0.008) | (0.017) |
| ERE2 | | 0.598 *** | | −0.0206 *** |
| | | (0.099) | | (0.003) |
| GDP | −0.037 | −0.042 | −0.0014 * | −0.0015 ** |
| | (0.027) | (0.027) | (0.001) | (0.001) |
| IS | −4.789 *** | −5.209 *** | 0.002 | −0.0123 |
| | (0.597) | (0.598) | (0.016) | (0.0162) |
| FDI | 5.655 *** | 4.401 *** | −0.121 *** | −0.165 *** |
| | (1.282) | (1.292) | (0.035) | (0.035) |
| TEL | −5.973 *** | −6.594 *** | 0.063 *** | 0.042 ** |
| | (0.609) | (0.614) | (0.017) | (0.017) |
| ROD | 0.323 *** | 0.324 *** | 0.007 *** | 0.007 *** |
| | (0.048) | (0.048) | (0.001) | (0.001) |
| ED | −1.056 * | −1.226 ** | 0.0632 *** | 0.057 *** |
| | (0.540) | (0.537) | (0.015) | (0.015) |
| RD | 1.583 *** | 1.620 *** | −0.022 ** | −0.021 ** |
| | (0.339) | (0.337) | (0.009) | (0.009) |
| Year | Control | Control | Control | Control |
| Reign | Control | Control | Control | Control |
| Constant | 43.61 *** | 48.53 *** | 0.493 *** | 0.662 *** |
| | (3.947) | (4.008) | (0.107) | (0.109) |
| Observations | 3024 | 3024 | 3024 | 3024 |
| R2 | 0.16 | 0.17 | 0.1 | 0.11 |

Note: The values in brackets are t-statistic values. ***, **, and * mean the values were shown to be significant at the levels of 1%, 5%, and 10% respectively.

## 5. Conclusions and Discussion

### 5.1. Summary and Conclusions

Based on the panel data of 216 prefecture-level cities in China from 2003 to 2016, five emission-reduction indicators were selected in this study (industrial $SO_2$ removal rate, soot removal rate, comprehensive utilization rate of industrial solid waste, domestic sewage treatment rate, and harmless treatment of domestic waste rate) to quantify the intensity of urban environmental regulations. Furthermore, the authors assessed the impact of environmental regulations on economic quality (green total factor productivity) and environmental quality (PM2.5) by observing the intensity of environmental regulations. The empirical results led us to the conclusions listed below:

(1) Considering resource consumption and pollution emissions, China's green total factor productivity is still increasing, and there is no situation in which the economic quality is improved at the expense of resources and the environment. (2) Nationwide, there is a U-shaped relationship between environmental regulations and the PM2.5 index, indicating that appropriate environmental regulations play a conducive role in developing environmental quality, while abating urban smog and excessive environmental regulations can cause damage to environmental quality. Otherwise, environmental regulations and green total factor productivity are in an inverted U-shaped correlation, revealing that appropriate environmental regulations can stimulate the improvement of urban green total factor productivity and economic quality, whereas if the intensity of environmental regulations becomes excessively strong, green total factor productivity suffers negatively instead of being increased. In sum, within a certain range, environmental regulations will positively affect the improvement of environmental quality and economic quality to achieve a win–win situation for environmental improvement and economic growth. (3) Considering different regions, the green total factor productivity of coastal areas is higher than that of inland areas, and the impact of environmental regulations on the economy and the environment is also different. On the one hand, while in coastal and inland areas environmental regulations have no nonlinear relationship with PM2.5, the passive impact of environmental regulations on PM2.5 in coastal cities is more significant than in inland areas, articulating that more emphasis should be placed upon strengthening environmental regulations in coastal cities. On the other hand, for coastal areas and inland areas, the impact of environmental regulations on green total factor productivity is an inverted U-shaped change, but the impact assessment results on total factor productivity in inland areas is higher than that in coastal areas. (4) R&D investment and education investment have not only rebounded to alleviate environmental issues such as smog, but also contribute to the improvement of green total factor productivity.

### 5.2. Suggestions and Enlightenment

(1) Advisable environmental regulations should be necessarily formulated by the government, and radical policies should be prevented. Based on the current situation, the government and relevant departments still should strengthen environmental regulations, improve the contribution of relevant laws and regulations, enhance supervision and punishment of environmental pollution to upgrade the economic quality by raising the green total factor productivity, and finally end at achieving a win–win situation of economy and environment, as expected.

(2) Additionally, despite those factors, the government is supposed to issue differentiated environmental policies according to local conditions. There are great distinctions in economic development levels and geographical location conditions among different regions. The environmental quality problem in coastal areas was revealed to be more serious than that in inland areas, though the level of economic development in coastal areas was higher than that in inland areas. The empirical results of this study illustrated that enhancing the intensity of environmental regulations can be more favorable to upgrading environmental quality in coastal areas. As the inverted U-shaped critical point of green total factor productivity was smaller than that of inland areas, policy-makers for coastal

regions should pay more attention to the intensity of local environmental regulations. Efforts should be made by government leaders to alleviate the haze problem step-by-step without further endangering the green total factor productivity.

(3) R&D (research and development) investments should be augmented to encourage the innovation and application of environmental-protection technologies. On the one hand, irrational behavior of and regulations issued by the government need sufficient scrutinizing, laying the foundation of prompting clean energy consumption by reforming the energy price mechanism. On the other hand, it is necessary to accelerate the promotion and application of cleaner production technologies, hasten production processes based on higher energy efficiency, and reach the final aim of reducing pollution emissions.

A multidimensional insight on the impact of the intensity of environmental regulations on the economy and the environment and its regional distinctions were proposed in this paper based on the panel data of 216 cities in China from 2003 to 2016. After measuring and analyzing, we reached some enlightening conclusions that enrich the theoretical and empirical research in related fields to a certain extent. However, due to the limitations of our research abilities, further studies are required to address some unsolved problems and imperfect parts. First of all, when we evaluated the results of affected economy and environment, we had not yet examined how the impact of environmental regulations on both will further affect the development of the entire society. Notwithstanding that the economy and the environment are two of the hotspots of social issues, they are simply important parts of this large society. Therefore, whether the "win–win" result of economy and environment generated from environmental regulations is beneficial to the realization of the goal of maximizing social welfare remains to be further explored in the future. Second, due to the difficulty of collection and shortage of data, this paper only selected urban data from 2003 to 2016. In follow-up research, the indicator calculation method should be further optimized, and more abundant data is expected to be collected for further improvement of the accuracy of the conclusions.

**Author Contributions:** Conceptualization, X.L. and B.C.; methodology, X.L.; software, X.L.; validation, X.L.; formal analysis, X.L. and B.C.; investigation, X.L.; resources, X.L.; data curation, X.L.; writing—original draft preparation, X.L.; writing—review and editing, Q.H.; visualization, X.L.; revisions and translations, Q.H. and C.X. All authors have read and agreed to the published version of the manuscript.

**Funding:** National Natural Science Foundation of China (72073012; 71873016).

**Institutional Review Board Statement:** Not applicable.

**Informed Consent Statement:** Not applicable.

**Data Availability Statement:** Data were selected from "China City Statistical Yearbook" (2004–2017) and "China Regional Economic Statistical Yearbook" (2004–2017) at https://www.cnki.net/ accessed on 19 May 2021.

**Conflicts of Interest:** The authors declare no conflict of interest.

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
