# Peer review of "Can a Win–Win Situation of Economy and Environment Be Achieved in Cities by the Government’s Environmental Regulations?"

_sustainability, doi:10.3390/su13115829_

Round 1
Reviewer 1 Report
1) An interesting term “Win-win Situation of Economy and Environment” of your title should be written first to make an interest to readers. Also, after reading your article, I recommend the following title for it:
Can a Win-win Situation of Economy and Environment Be Achieved in Cities by the Government’s Environmental Regulations?
2) In line 10, look at your sentence ‘This paper selects city-level panel data from 216 cities in my country from 2003 to 2016.’ You are all four authors―must be ‘in our country’, not ‘in my country’; though, ‘in China’ is better.
3) In general, your English writing styles are needed proofreading; in particular, the Abstract is not easier for readers to understand.
4) In line 27, for Section 1 ‘Introduction’ is enough because the Introduction section usually includes Background; it is also aligning with the Journal’s suggested structure.
5) In lines 117-118, your first research question ‘1. Although environmental regulation has an impact on local economy and environment, is it all positive?’ is awkward, not interestingly presented. Is it okay if you present it like ‘What are the positive and negative impacts of environmental regulations on the local economy and environment’?
6) Your third research question ‘Whether there are regional differences in the impact of environmental regulation on the economy and environment due to different economic development and pollution degree in different regions’ seems not a real question?
7) Denoting the structure of the paper in lines 125-132 is not necessary and should be deleted.
8) In line 160, provide what the GMM stands for.
9) In line 202, provide what the TFP stands for.
10) I did not see you pointed out Figure 1 in the text.
11) In line 232, my country again; see comment 2).
12) I would like to see a bit more detail on how you selected the control variables in Table 1. I suggest using a table to show where each variable came from with denoting reference no. in the same row (see samples in reference a below―Table A1). Also, please identify the variable dimensions (see samples in reference b below―Table 5), but your research is only economic and environmental dimensions.
- Table A1 (Appendix)―Gwiaździńska-Goraj, M.; Pawlewicz, K.; Jezierska-Thöle, A. Differences in the Quantitative Demographic Potential—A Comparative Study of Polish–German and Polish–Lithuanian Transborder Regions. Sustainability2020, 12, 9414. https://doi.org/10.3390/su12229414
- Table 5―Chan, P. Assessing Sustainability of the Capital and Emerging Secondary Cities of Cambodia Based on the 2018 Commune Database. Data2020, 5, 79. https://doi.org/10.3390/data5030079
13) In general, I am curious why your research focused only on economic and environmental, not on social? Without discussing sustainability issues, how can it fit the Sustainability journal? As your research is very potential to discuss the sustainability aspect, I believe that if you discuss it, your research will be more applicable and interesting.
Reviewer 2 Report
I would like to congratulate the authors for the comprehensive work done within this article. It was a pleasure to follow each section: very well argued both technically and economically.
My only recommendation would be a more concise outline of the main policy implications and the addition of a paragraph on future research directions.
Reviewer 3 Report
The authors of the paper illustrate the influence of environmental regulations on environmental quality and economic quality 11 based on the green total factor productivity.
The subject of the paper is good. However the style of the paper has to be improved.
The novelty of the paper should be highlighted more clearly in all sections.
ML and U-Shaped are abbreviations, highlight that in the abstract and introduction.
The presented figures are very poor. The authors have to improve them.
Each table has to be presented same page without dividing them.
The introduction section is weak, the motivation, justification, and contribution should be more emphasizes in the introduction section. Moreover, it suggest presenting some related works about public transportation in Line 108.
I suggest the following reflection points:
- Gündoğdu, F. K., Duleba, S., Moslem, S., & Aydın, S. (2021). Evaluating public transport service quality using picture fuzzy analytic hierarchy process and linear assignment model. Applied Soft Computing, 100, 106920.
- Duleba, S., Mishina, T., & Shimazaki, Y. (2012). A dynamic analysis on public bus transport's supply quality by using AHP. Transport, 27(3), 268-275.
- Moslem, S., & Çelikbilek, Y. (2020). An integrated grey AHP-MOORA model for ameliorating public transport service quality. European Transport Research Review, 12(1), 1-13.
Round 2
Reviewer 1 Report
The authors took all my comments into account. I have no further comments, but please check the following errors:
- Seems like the authors misused the track-change function that made confusion between the old and revised versions. So please check all the text again.
- The title of Section 1 "Introduction" is enough.
- There are so many text errors in Figure 1.
- The title of Section 5 "Conclusions and Discussion" is enough. And 5.1 should be entitled "Summary and Conclusions".
Reviewer 3 Report
There are several mistakes and the paper is improved, however, not carefully.
Line 33. correct the title.
Line 128. many errors like the end of this line.
Figure 1. is not clear, present is in different way.
Table 1. has to be on the one page.
Line 905. 2 references are mixed
Round 3
Reviewer 3 Report
All points have been improved.
I recommend English proofreading.